# Kill Two Birds with One Stone? The Effect of *Helicobacter pylori* Eradication in Decreased Prevalence of Gastric Cancer and Colorectal Cancer

**DOI:** 10.3390/cancers16223881

**Published:** 2024-11-20

**Authors:** Yang-Che Kuo, Hung-Ju Ko, Lo-Yip Yu, Shou-Chuan Shih, Horng-Yuan Wang, Ying-Chun Lin, Kuang-Chun Hu

**Affiliations:** 1Division of Gastroenterology, Department of Internal Medicine, MacKay Memorial Hospital, Taipei 104217, Taiwan; kuoyangche@yahoo.com.tw (Y.-C.K.); benny7190@gmail.com (L.-Y.Y.); shihshou@gmail.com (S.-C.S.); mmh4013@gmail.com (H.-Y.W.); 2Healthy Evaluation Center, MacKay Memorial Hospital, Taipei 104217, Taiwan; bonnie@mmh.org.tw; 3MacKay Junior College of Medicine, Nursing and Management, Taipei 112021, Taiwan; 4Department of Anesthesiology, MacKay Memorial Hospital, Taipei 104217, Taiwan; elegant.beaver@gmail.com; 5Graduate Institute of Epidemiology and Preventive Medicine, College of Public Health, National Taiwan University, Taipei 106319, Taiwan; 6MacKay Medical College, Taipei 252005, Taiwan

**Keywords:** *H. pylori* infection, colorectal cancer, gastric cancer

## Abstract

We reviewed the relationship of *Helicobacter pylori* infection with gastric adenocarcinoma and colorectal neoplasm/adenocarcinoma. The potential difference in the hazard ratio between untreated and treated *H. pylori* infection in relation to colorectal cancer risk may be underestimated. We tried to find unmet issues about *H. pylori* infection connected to colorectal adenocarcinoma formation. We are also putting forward potential research objectives to investigate the mechanism by which *H. pylori* infection may contribute to an increased incidence of colorectal cancer.

## 1. Introduction

In humans, the relationship between microbial infections and tumor formation is well established. For example, *Helicobacter pylori* (*H. pylori*) is linked to GC, hepatitis B and/or hepatitis C viruses are associated with hepatocellular carcinoma, and human papillomavirus (HPV) is implicated in the development of cervical cancer [1]. Thirty years ago, researchers identified a potential link between *H. pylori* infection and the development of colorectal neoplasms. However, at the time, the association was not definitively established due to the absence of large-scale retrospective cohort studies needed to confirm the connection. In a recent study on *H. pylori*-positive individuals, it was found that those who underwent *H. pylori* eradication treatment experienced a significant reduction in both CRC incidence and mortality compared to those who did not receive treatment [2]. This finding suggests that *H. pylori* infection may be associated with the development of both GC and CRC, highlighting the need for further investigation and evaluation.

## 2. Prevalence of *H. pylori* Infection in the World

The global prevalence of *H. pylori* infection varies widely by region, with higher rates observed in developing countries compared to developed countries. In general, the prevalence is higher in older age groups, lower socioeconomic status, and in individuals living in crowded or unsanitary conditions. Factors such as poor sanitation, the lack of access to clean water, and overcrowding contribute to the transmission of *H. pylori* [3]. In developing countries, the prevalence of *H. pylori* infection can be as high as 80–90% in some populations, particularly in rural areas with poor living conditions [4]. In contrast, the prevalence in developed countries is generally lower, ranging from 20 to 50%. However, there are significant variations within countries and regions, influenced by factors such as ethnicity, socioeconomic status, and living conditions [5].

In Asia, particularly in countries like China, India, and Bangladesh, the prevalence of *H. pylori* infection is high, with some studies reporting rates exceeding 50–60% in certain populations. Similarly, in Africa, high prevalence rates have been reported, especially in sub-Saharan regions. In Latin America, the prevalence is also considerable, with some countries showing rates as high as 70–80% in certain communities [5,6]. In Europe and North America, the prevalence of *H. pylori* infection is generally lower compared to developing regions, but there are still significant variations within countries. For example, studies have shown higher prevalence rates among certain ethnic groups and immigrant populations [7].

The prevalence of *H. pylori* infection is not only influenced by geographical location but also by individual risk factors such as age, socioeconomic status, and lifestyle. Older age is consistently associated with higher prevalence rates, likely due to cumulative exposure over time. Lower socioeconomic status and poor living conditions contribute to the increased transmission of *H. pylori*, as seen in many developing countries [8].

The impact of *H. pylori* infection on public health cannot be overstated, given its association with chronic gastritis, peptic ulcers, and GC. As such, understanding its prevalence and risk factors is crucial for developing targeted interventions to reduce its burden on global health. Efforts to control and prevent *H. pylori* infection should address both individual risk factors and broader public health measures to improve living conditions and reduce transmission. The potential advantages of *H. pylori* eradication include the following: reducing ulcer risk, potential cancer prevention to gastric cancer and colon cancer, symptom relief to patient with gastritis or dysplasia, and improving quality of life. At the same time, antibiotic treatment may induce some side effects in patients, like nausea, diarrhea, and abdominal pain, especially for some people who must undergo complicated and multiple eradication therapy for antibiotic resistance. Understanding the global burden of *H. pylori* infection is essential for guiding effective strategies to mitigate its impact on public health.

## 3. *H. pylori* Infection and Gastric Malignancy Disease

In 1982, Warren and Marshal discovered a connection between *H. pylori* and gastric ulcer disease [9]. Since then, this bacterium has been the subject of many studies in the field of gastroenterology. The link between *H. pylori* infection and GC is extensively studied, and it is now widely accepted that *H. pylori* plays a significant role in the pathogenesis of this malignancy. The bacterium is believed to contribute to the development of GC through several mechanisms, including the induction of chronic inflammation, the production of virulence factors that damage the gastric mucosa, and the alteration of host cell signaling pathways.

Chronic inflammation is a key factor in the development of cancer, and *H. pylori* infection is known to induce a persistent inflammatory response in the gastric mucosa. This chronic inflammation can lead to the accumulation of genetic mutations, DNA damage, and ultimately the transformation of normal gastric epithelial cells into cancerous cells. Additionally, *H. pylori* produces virulence factors such as the cytotoxin-associated gene A (CagA) and vacuolating cytotoxin A (VacA) that directly contribute to the damage of gastric epithelial cells, further promoting carcinogenesis [10,11].

Furthermore, *H. pylori* infection has been shown to disrupt host cell signaling pathways involved in cell proliferation, apoptosis, and DNA repair. The CagA protein is delivered into gastric epithelial cells by a bacterial type IV secretion system. CagA interacts with various host cell proteins, including the prooncogenic phosphatase SH2-domain-containing protein tyrosine phosphatase (SHP2), leading to an abnormal activation of the RAS–ERK pathway. This abnormal activation is implicated in *H. pylori*-mediated gastric carcinogenesis, which is associated with genomic alterations in the host cell. *H. pylori* infection has been shown to induce DNA double-stranded breaks (DSBs) through both CagA-dependent and CagA-independent mechanisms. Furthermore, *H. pylori* also interferes with multiple DNA damage responses and DNA repair systems. By interfering with these critical cellular processes, *H. pylori* can promote the uncontrolled growth and survival of damaged cells, increasing the risk of malignant transformation (Figure 1) [12,13].

*H. pylori* may activate Nuclear Factor-κB (NF-κB) through classical or alternative pathways, depending on the cell type. In gastric epithelial cells, the activation of NF-κB by *H. pylori* is mainly through the classical pathway, which is dependent on the Cag pathogenicity island (Cag PAI). In this pathway, CagA can physically associate with and enhance the activity of TAK1, which then activates NF-κB through the Tumor necrosis factor receptor-associated factor 6 (TRAF6)-mediated, Lys 63-linked ubiquitination of TGFβ activated kinase (TAK1). Additionally, the CagA-dependent activation of NF-κB may also be mediated through the CagA-induced activation of phosphatidylinositol 3-kinase (PI3K)/AKT signaling, which in turns activates the inhibitor of NF-κB (IκB) kinase (IKK) directly or indirectly via TAK1. The role of PI3K signaling in the CagA-induced NF-κB activation and inflammation was supported by the observation that either PI3K inhibition or a small interfering RNA (siRNA) mediated knockdown of NF-κB p65 could suppress *H. pylori*-induced interleukin-8 (IL-8) expression. The activation of NF-κB and up-regulation of IL-8 in gastric epithelial cells were proposed as the critical mechanisms underlying *H. pylori*-induced chronic inflammation and gastric carcinogenesis (Table 1) [14].

In addition, the International Agency for Research on Cancer recognized *H. pylori* as a class I carcinogen in 1994 [15]. A previous study demonstrated that individuals infected with *H. pylori* had a significantly higher risk of developing noncardiac gastric cancers (GCs), with a sixfold increase in risk compared to uninfected individuals (odd ratio [OR]: 5.9; 95% confidence interval [CI]: 3.4–10.3) (Table 1) [16]. Chiang et al. confirmed the long-term benefits of *H. pylori* eradication, showing a 53% reduction in the occurrence of GC among the high-risk population [17]. Based on this evidence, all individuals infected with *H. pylori* are recommended to be offered eradication therapy. Vulnerable individuals should be tested for the presence of this infection and treated if the test result is positive, especially in individuals at a high risk of developing GC [18].

The mechanism of *H. pylori*-induced GC formation has been well studied. The very low pH level in the stomach environment inhibits the survival of general microorganisms. Nevertheless, with the help of ammonia produced by urease, *H. pylori* is able to counteract the acidic conditions in the cytosol, periplasm, and on its surface within the stomach’s harsh environment [19]. This challenging environment may contribute to the dominance of *H. pylori* in the stomach. In gastric epithelial cells, the presence of a functional Cag PAI in *H. pylori* resulted in the observed cell scattering effects caused by cytoskeletal modifications and pro-inflammatory responses triggered by the transcription factor NF-κB [20,21]. Additionally, the activation of growth factor receptors, cell proliferation, inhibition of apoptosis, invasion, and angiogenesis were also observed through Cag A [22] (Figure 1).

The association between *H. pylori* infection and GC has significant implications for public health, particularly in regions with a high prevalence of both *H. pylori* infection and GC. Efforts to eradicate *H. pylori* through antibiotic therapy have been explored as a potential strategy for preventing GC. However, the widespread use of antibiotics for this purpose raises concerns about antimicrobial resistance and the potential for unintended consequences on the gut microbiota.

## 4. *H. pylori* Infection and Colorectal Neoplasm/Adenocarcinoma

Since the 1990s, the association between *H. pylori* and the development of colorectal neoplasms has been extensively debated among researchers. *H. pylori* has been reported to be linked to both colorectal neoplasm and/or adenocarcinoma growths in the colon. For example, *H. pylori* reportedly contribute to a 1.3- to 1.97-fold increased risk of colon adenoma, with or without high-grade dysplasia [23,24]. However, some studies disagreed due to their findings of a minimal increase in the incidence of colon adenoma in patients with *H. pylori* infection [25,26], although two meta-analyses revealed a significant and positive association between *H. pylori* infection and the risk of colorectal adenoma (OR: 1.49, 95% CI: 1.37–1.62) and CRC (OR: 1.70, 95% CI: 1.64–1.76) (Table 1) [27,28]. Nevertheless, these studies that only investigated the link between *H. pylori* infection and colorectal adenoma have not thoroughly discussed the cause and effect due to their use of cross-sectional study designs.

Our previous research findings indicated that individuals with persistent *H. pylori* positive test results had a significantly higher risk of developing colorectal adenoma than those with *H. pylori* negative test results (HR: 3.34, 95% CI, 1.899–5.864) [29]. However, the connection between *H. pylori* and CRC was not included. Recently, Shah et al.’s study conducted in the United States found that individuals testing positive for *H. pylori* had 18% (adjusted hazard ratio [HR]: 1.18, 95% CI, 1.12–1.24) and 12% (adjusted HR: 1.12, 95% CI, 1.03–1.21) higher risks of developing incident CRC and fatal CRC, respectively, compared to those testing negative. Additionally, the treatment for *H. pylori* was associated with a 0.23–0.35% absolute risk reduction in incident and fatal CRCs. These findings highlight the potential impact of *H. pylori* infection and its treatment on the risk of developing CRC [2]. The authors also analyzed various racial populations, including White, Black, Asian/Pacific Islander, American Indian, and Hispanic groups, demonstrating that *H. pylori* infection may influence the development of CRC, thereby strengthening the study’s findings. Based on current knowledge, a clinical trial specifically focused on investigating the potential reduction in the incidence of CRC through *H. pylori* treatment is lacking.

However, Epplein et al. also pointed out more important issues. Their study was not sufficiently powered to effectively investigate the correlation between successful *H. pylori* eradication and treatment [30]. In Shah et al.’s study, the participants testing positive for *H. pylori* were categorized into the treated and untreated groups and whether they achieved successful eradication or not. Due to wide use of antibiotics in the past decades, antibiotic resistance among patients with an *H. pylori* infection had become an important issue. In Hong et al.’s study, the primary antibiotic resistance rates of *H. pylori* were as follows: 22% for clarithromycin, 52% for metronidazole, 26% for levofloxacin, 4% for tetracycline, and 4% for amoxicillin [31]. This result indicates that, in the *H. pylori*-treated group, approximately 20–25% of the patients still had this bacterial infection, affecting the study results. The hazard ratio difference between the untreated and treated *H. pylori* infection cases in the risks for incident and fatal CRCs might have been underestimated. Epplein et al. also emphasized the importance of retesting the patients to ensure the complete eradication of the bacteria and highlighted the ongoing necessity of understanding the reasons and preventive measures for treatment failure [30]. This research is particularly crucial as we work toward implementing the *H. pylori* test-and-treat approach on a larger scale.

## 5. Unmet Issues About *H. pylori* Infection and Colorectal Adenocarcinoma

Plummer raised a question regarding the discrepancy between the high prevalence of *H. pylori* infections in certain areas and the low risk of CRC in these areas [32]. For this query, Parkin’s study revealed a higher prevalence of *H. pylori* infections but a lower incidence of colorectal adenoma and CRC in specific regions, such as most parts of Asia, some Eastern European countries, and selected countries in South America. Interestingly, these areas also exhibited a low prevalence of diabetes mellitus (DM) [33,34]. These findings suggest that the development of colorectal adenoma may be influenced by multiple factors, with *H. pylori* infection being one of them. It is also possible that an increase in DM prevalence in certain areas could be associated with an increase in the incidence of colorectal adenoma.

Several studies demonstrated that *H. pylori* infection might contribute to the development of hyperglycemia and DM. One potential pathway involves the initiation of persistent low-grade inflammation. *H. pylori* infection is recognized for eliciting an inflammatory reaction in the gastric mucosa, causing the secretion of pro-inflammatory cytokines like tumor necrosis factor-alpha (TNF-α) and interleukin-6 (IL-6). These cytokines have the capacity to induce insulin resistance, a critical characteristic of type 2 DM, by disrupting insulin signaling pathways in organs such as the liver, muscles, and adipose tissue [35]. In addition to promoting inflammation, *H. pylori* infection may also alter the gut microbiota, leading to dysbiosis. Dysbiosis refers to an imbalance in the composition and function of the gut microbiota, which has been implicated in the pathogenesis of various metabolic disorders, including DM. Studies have shown that *H. pylori* infection is associated with changes in the gut microbiota, including a decrease in microbial diversity and an increase in potentially pathogenic bacteria. These alterations in the gut microbiota may contribute to the development of DM by promoting systemic inflammation and metabolic dysfunction [36]. *H. pylori* infection also has been linked to the dysregulation of gut hormones involved in the regulation of glucose metabolism. For example, ghrelin, a hormone produced by the stomach that stimulates appetite and promotes insulin secretion, has been found to be elevated in individuals with *H. pylori* infection. The dysregulation of ghrelin signaling may disrupt glucose homeostasis and contribute to the development of DM [37].

A combination of both *H. pylori* infection and hyperglycemia status might increase colorectal neoplasm formation. Previous systemic review research revealed a significant correlation between the prevalence of DM in a population and the risk of developing colorectal adenoma in the presence of *H. pylori* infection. With a DM prevalence exceeding 6% in the study population, our study underscores the heightened impact of *H. pylori* infection in the development of colorectal adenoma [38]. This result might explain why some areas had a high prevalence of *H. pylori* infections but a low incidence of CRC. It also hints that, when the DM prevalence increases in these areas, the incidence of colorectal neoplasm might also increase. Our prior investigation showed that the combination of high blood glucose levels and *H. pylori* infection played a role in the development of colon adenomas and had a synergistic impact [39]. Kim et al. also demonstrated that the presence of *H. pylori* infection was significantly associated with the risk of developing colorectal adenomas [40]. We suggested that they recheck their database and consider adding the effects of hyperglycemia in their study [41]. They found that participants with both *H. pylori* infection and hyperglycemia had the highest risk of developing any adenoma (OR: 1.61; 95% CI, 1.09–2.37) and advanced neoplasia (OR: 2.01; 95% CI, 1.16–3.47). They also found similar results that showed the combination of *H. pylori* infection and hyperglycemia has a synergistic effect on the risk of developing colorectal neoplasia [42].

Our previous research also revealed that, even when the Waist-to-Hip ratio (WHR) is 0.9, individuals with an elevated Framingham Risk Score (FRS), high HbA1c levels, or a positive *H. pylori* infection still require careful monitoring for colorectal adenoma risk. These factors have an interdependent relationship, and, when considering the combined effects of the WHR and the FRS, there is only a minimal reduction in adenoma risk [43]. Considering that tumor cell formation was induced by multiple factors, whether *H. pylori* infection status, diabetes, and/or smoking influence the development of CRC formation needs to be further evaluated.

## 6. Potential Impact of *H. pylori* Infection in Colorectal Neoplasm Formation

The potential mechanisms of the causal relationship between *H. pylori* infection and CRC might show the direct and/or indirect effects of *H. pylori* on colorectal carcinogenesis [44]. To produce a direct effect, the bacterium or molecular effectors, such as secreted toxins, must be specifically located within the colorectal tissue. We had previously attempted to collect blood gut flora metabolites, such as trimethylamine-N-oxide and endotoxins, from our patient to examine their possible linkage with carotid artery plaque or colon neoplasm formation. Unfortunately, we failed to find a connection between these two gut-flora metabolites and carotid artery plaque or colorectal adenoma [45]. According to Epplein and Julia et al.’s perspective, *H. pylori* infections may indirectly contribute to the formation of colorectal neoplasms through several potential pathways [44]. We think that the possible research directions that are more likely to lead to understanding why *H. pylori* eradication would decrease the CRC incidence are as follows: First, the persistent existence of *H. pylori* might induce the capture of lipopolysaccharide by gut mucosa toll-like receptors, leading to a subsequent increase in the IL-23 levels, which acts on downstream cells, including lymphocytes. This action triggers an inflammatory process that activates the signal transducer and activator of the transcription 3 (STAT 3) pathway. Consequently, this pathway promotes cell proliferation and survival, ultimately leading to tumorigenesis (Figure 1) (Table 1) [46].

Furthermore, emerging evidence suggests that *H. pylori* infections may modulate host immune responses and alter immune surveillance mechanisms in the colonic mucosa, thereby facilitating the escape of neoplastic cells from immune recognition and clearance. This immune evasion strategy could potentially contribute to the initiation and progression of colorectal neoplasms in individuals infected with *H. pylori*. Recently, a study also demonstrated that *H. pylori* infections accelerated tumor development in *Apc*-mutant mice. Ralser A. et al. identified a distinct *H. pylori*-driven immune alteration signature characterized by a decrease in regulatory T cells and an increase in pro-inflammatory T cells. Additionally, within the intestinal and colonic epithelium, *H. pylori* has been shown to induce pro-carcinogenic STAT3 signaling and a reduction in goblet cells. These changes, when combined with pro-inflammatory and mucus-degrading microbial signatures, have been linked to tumor development [47]. Furthermore, when *Apc*-mutant mice were housed under germ-free conditions, the incidence of tumors was reduced. Genua et al.’s study showed the correlation between *H. pylori* antigen levels, and the risk of colorectal neoplasia decreases as adenoma progression occurs, indicating the early onset of the immune response at the polyp stage [48]. Additionally, the early antibiotic eradication of *H. pylori* infection normalized the tumor incidence to the level of uninfected controls (Figure 1).

Second, *H. pylori* is not the only microorganism related to CRC that is present in the gut microbiota. Heimesaat et al. conducted a thorough investigation into the changes in microbiota throughout the gastrointestinal tract of Mongolian gerbils following 14 months of *H. pylori* infection. The study revealed significant alterations in the composition of microbiota within the large intestine, accompanied by a notable pro-inflammatory response characterized by heightened CD3+ T-cell infiltration in the colonic tissue. Notably, the presence of the mucus-degrading species *Akkermansia* was observed in the cecum and colon of *H. pylori*-infected mice, indicating its potential contribution to compromised intestinal barrier function. Furthermore, elevated levels of *E. coli, Enterococcus* spp., and *Bacteroides/Prevotella* spp. were identified in *H. pylori*-infected gerbils compared to non-infected control subjects. These findings shed light on the intricate interplay between *H. pylori* infection and the host microbiota, providing valuable insights into the pathophysiological mechanisms underlying gastrointestinal tract alterations associated with this prevalent bacterial infection [49].

Following *H. pylori* eradication, several studies have documented shifts in gut microbiota composition. Research indicates that the elimination of *H. pylori* can lead to an increase in certain beneficial bacterial populations while simultaneously decreasing others that may have been previously suppressed. First, after *H. pylori* eradication, the host colon microbiota might increase diversity. Many individuals exhibit an increase in microbial diversity in the colon after *H. pylori* eradication. This diversity is often associated with improved gut health and resilience against pathogenic organisms. Second, the gut bacterial populations might alert specific bacterial genera such as *Bacteroidetes* to decrease and *Firmicutes* to increase during long-term follow-up after *H. pylori* eradication [50,51]. Third, changes in gut microbiota composition can have various health implications. For example, an increase in beneficial bacteria may enhance gut barrier function and immune responses, while a decrease in protective species could predispose individuals to gastrointestinal disorders [52].

Wong et al. demonstrated through sequencing that there are significant shifts in microbial composition and ecology in individuals with CRC. Additionally, research has identified specific bacteria, such as *Fusobacterium nucleatum,* certain strains of *Escherichia coli*, *Bacteroides fragilis*, and their respective roles in the development of CRC. Especially, *Streptococcus gallolyticus, Bacteroides fragilis,* and *Fusobacterium nucleatum* could have direct effects on the formation of colon neoplasms or cancer [1]. Several human shotgun metagenomic and 16 S ribosomal RNA sequencing studies were conducted to analyze the microbiota associated with CRC, using fecal and mucosal samples to characterize these microorganisms. *H. pylori* infection has been linked to notable changes in the gut microbiota composition. A study revealed that thirteen specific taxa exhibited significant differences in abundance between *H. pylori*-positive individuals and negative controls. Notably, there was an enrichment of *Prevotellaceae* abundance in patients with *H. pylori* infection. *Prevotellaceae* is recognized as a pro-carcinogenic taxa, further emphasizing the potential implications of *H. pylori* infection on gastrointestinal health. These findings underscore the intricate interplay between *H. pylori* and the gut microbiota, shedding light on potential avenues for further research and therapeutic interventions in the field of gastroenterology [53].

Recently, Lee et al. conducted a study in which participants were randomized to either an invitation for an *H. pylori* stool antigen (HPSA) combined with a fecal immunochemical test (FIT) assessment or an FIT alone. They found that patients who accepted an invitation for an HPSA-combined FIT were associated with lower rates of GC (0.79 [95% CI, 0.63–0.98]) but not with GC mortality (1.02 [95% CI, 0.73–1.40]), compared with an FIT alone [54]. This finding means that using stool examination could screen not only CRC but also GC and might decrease these two malignant diseases’ probability. Shah et al. showed that *H. pylori* infection may be linked to a slightly increased risk of developing CRC and higher mortality rates [2]. This finding suggests that *H. pylori* may play a role in the development of colon neoplasms by serving as a “biomarker” or an “indicator organism” that indicates exposure to immune-stimulating carcinogenic bacteria or antigens (Table 1). *H. pylori* may alter the gut microbiota composition and function, leading to dysbiosis and the subsequent promotion of colorectal neoplasm formation. The eradication of this bacterium may or may not serve as an indicator of potential changes in the overall composition of the human microbiota [55]. Compared to the laboratory tests performed for cases of *S. gallolyticus, B. fragilis,* and *F. nucleatum* infections, using the biopsy urease or urea breath test was a more convenient and onco-economic method for detecting the presence of *H. pylori.*

## 7. Conclusions

From a clinical perspective, the potential impact of *H. pylori* infection on colorectal neoplasm formation underscores the importance of considering *H. pylori* eradication therapy not only for its established role in preventing peptic ulcers and GC but also for its potential impact on colorectal health. However, it is essential to weigh the potential benefits of *H. pylori* eradication against the risks and costs associated with antibiotic treatment, particularly in populations with a low prevalence of GC and a high incidence of antibiotic resistance. In conclusion, more research involving animal models, studies with biomarker examinations, and case–control studies are necessary in the future to conduct a comprehensive investigation into the detailed mechanisms of *H. pylori* related to CRC formation. The discovery of the detailed process of *H. pylori* infection would significantly highlight the importance of eradicating this bacterium in the field of public health. Given that *H. pylori* infection is considered the most infectious disease in humankind, understanding its intricate process would further emphasize the urgency of addressing this public health concern. This research could present an opportunity to kill two birds (GC and CRC) with one stone (*H. pylori* eradication).

**Table 1 cancers-16-03881-t001:** *H. pylori* effect in gastric adenocarcinoma and colorectal adenocarcinoma.

	Gastric Adenocarcinoma	Colorectal Adenocarcinoma
*H. pylori* odd ratio for cancer	5.9 [16]	1.7 [27,28]
95% confidence interval	3.4–10.3 [16]	1.64–1.76 [27,28]
Decreased cancer incidence relative risk after *H. pylori* eradication	53% reduction	31% reduction
Mechanism	Low pH in the stomach reduced other bacterium and *H. pylori* become pre-dominated *Cag* PAI induced cell scattering [14] CagA-deregulated SHP2 hyperstimulates RAS-ERK signaling [12,13] NF-κB trigger pro-inflammatory [14]	*H. pylori* leading to increasing IL-23 and triggers an inflammatory process that activates the STAT-3 [46] A “biomarker” or an “indicator organism” indicates exposure to immune-stimulating carcinogenic bacteria [55]

*H. pylori*: *Helicobacter pylori*; *Cag* PAI: cag pathogenicity island; STAT-3: signal transducer and activator of the transcription 3; SHP2: SH2-domain-containing protein tyrosine phosphatase; and NF-κB: nuclear factor-κB.

## Figures and Tables

**Figure 1 cancers-16-03881-f001:**
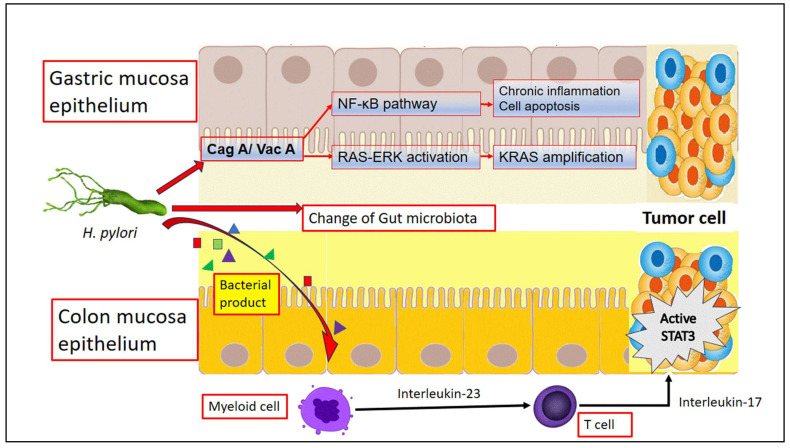
Possible mechanism of *H. pylori* effect in gastric and colon tumor cell formation. SAT-3: signal transducer and activator of the transcription 3; CagA: cytotoxin-associated gene A; and VacA: vacuolating cytotoxin A.

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
