# Peer review of "Kill Two Birds with One Stone? The Effect of Helicobacter pylori Eradication in Decreased Prevalence of Gastric Cancer and Colorectal Cancer"

_cancers, 2024, doi:10.3390/cancers16223881_

Round 1
Reviewer 1 Report
Comments and Suggestions for Authors
The reviewed manuscript explores the link between Helicobacter pylori (H. pylori) infection and its potential role in the development of gastric cancer (GC) and colorectal cancer (CRC). The review emphasizes that while the association between H. pylori and GC is well-established, the connection with CRC remains under debate. Retrospective studies suggest that treatment for H. pylori infection may reduce CRC incidence, but further research is needed to clarify the underlying mechanisms. The paper discusses the global prevalence of H. pylori and its association with gastrointestinal diseases, including peptic ulcers and chronic gastritis. Additionally, eradication therapy shows promise, though antibiotic resistance poses challenges.
The review is thorough in covering the relationship between H. pylori infection and gastrointestinal malignancies, but suffers from some structural and formatting issues that detract from its clarity. Suggestions include improved definitions and terminology consistency (such as abbreviations like GC and CRC), providing full names for key terms (e.g., PAI, TAK1), and ensuring correct citation formatting. These revisions will improve the manuscript’s flow and readability.
Specific Comments:
Lines 54-58: I suggest rewriting this part to: “In humans, the relationship between microbial infections and tumor formation is well established. For example, Helicobacter pylori (H. pylori) is linked to gastric cancer, hepatitis B and/or hepatitis C viruses are associated with hepatocellular carcinoma and human papillomavirus (HPV) is implicated in the development of cervical cancer. Thirty years ago, researchers identified a potential link between H. pylori infection and the development of colorectal neoplasms. However, at the time, the association was not definitively established due to the absence of large-scale retrospective cohort studies needed to confirm the connection.”
Lines 61-63: I suggest rewriting this to: “This suggests that H. pylori infection may be associated with the development of both GC and CRC, highlighting the need for further investigation and evaluation”.
Line 71: H.pylori should be in italics review the whole manuscript.
Line 92, 105: “gastric cancer” should be shortened to “GC” according to the first abbreviation appearing in the text. Multiple such errors have been detected in the manuscript. Please review and correct them accordingly.
Line 101: space is lacking before [9].
Line 117: “Type IV” should be “type IV”
Line 120: “Ras–ERK” should be “RAS-ERK”
Line 126: Should be (Figure 1/2) – provide figure 1 after mentioning it in the manuscript.
Line 127: NF-κB should be provided with a full name.
Line 129: What is PAI. Provide full name.
Line 130: TAK1, TRAF6 lacks full name the first time it appears in the manuscript.
Line 133: “Akt” should be replaced with “AKT”
Line 137: IL – need the full name.
Line 141, 144 – lack of space before citation [x] – multiple such errors have been detected in the manuscript. Please review and correct them accordingly.
Line 173-175: Please rewrite: “However, some studies disagreed due to their findings of a minimal increase in the incidence of colon adenoma in patients with H. pylori infection” for enhanced clarity and precision.
Line 177: introduces the new abbreviation “colorectal cancer (CRC)” which appeared multiple times in the text before the first introduction. Please decide which one to use – the abbreviated form or the full form throughout the manuscript.
Line 191-193: “The authors also examined white, black, Asian/Pacific Islander, American, Indian, and Hispanic race populations and showed that H. pylori infection might affect the development of CRC and let this study result was more consolidated” Please rewrite for clarity.
Lines 262-265: Please rewrite for clarity.
Line 310: Remove “in animal study”
Line 336: Please rewrite from: “Wong et al. had demonstrated, via sequencing, uncovered shifts in microbial composition and ecology in individuals with CRC” to “Wong et al. demonstrated through sequencing that there are significant shifts in microbial composition and ecology in individuals with CRC”
Line 337-341: “Additionally, research has identified specific bacteria, such as Fusobacterium nucleatum, certain strains of Escherichia coli, and Bacteroides fragilis, and their respective roles in the development of CRC. Especially, Streptococcus gallolyticus, B. fragilis, and F. nucleatum could have direct effects on the formation of colon neoplasms or cancer” please decide if you want to provide a full name or abbreviation e.g. Bacteroides fragilis or B. fragilis please unify it throughout the manuscript.
Line 341: Consider rewriting to “Several human shotgun metagenomic and 16S ribosomal RNA sequencing studies have been conducted to analyze the microbiota associated with colorectal cancer (CRC), using fecal and mucosal samples to characterize these microorganisms.”
Table 1. “odd ratio” seems highlighted.
Line 379: “Since H. pylori infection was the most infectious disease in human kind.” This statement seems unfinished or oddly inserted here. Please rewrite.
The reference list seems disordered please correct and check whether the reference numbers match the content of the manuscript. Provide a reference list and the statements (including contribution statements) according to the journal requirements. Pay attention to the layout requirements of the journal - including affiliations.
.
Comments on the Quality of English Language
Minor editing of English language required.
Author Response
|
Comments 1: Lines 54-58: I suggest rewriting this part to: “In humans, the relationship between microbial infections and tumor formation is well established. For example, Helicobacter pylori (H. pylori) is linked to gastric cancer, hepatitis B and/or hepatitis C viruses are associated with hepatocellular carcinoma and human papillomavirus (HPV) is implicated in the development of cervical cancer. Thirty years ago, researchers identified a potential link between H. pylori infection and the development of colorectal neoplasms. However, at the time, the association was not definitively established due to the absence of large-scale retrospective cohort studies needed to confirm the connection.” |
|
Response 1: Thank you for pointing this out. We agree with this comment. Therefore, we had rewritten this section as your suggestion in line 54-61.
|
|
Comments 2: Lines 61-63: I suggest rewriting this to: “This suggests that H. pylori infection may be associated with the development of both GC and CRC, highlighting the need for further investigation and evaluation”. |
|
Response 2: Agree. Thank you for your suggestion. We had rewritten this section as your suggestion in line 64-65.
|
|
Comments 3: Line 71: H.pylori should be in italics review the whole manuscript. Response 3: Thanks for your suggestion and we had changed this word in the whole manuscript.
Comments 4: Line 92, 105: “gastric cancer” should be shortened to “GC” according to the first abbreviation appearing in the text. Multiple such errors have been detected in the manuscript. Please review and correct them accordingly. Response 4: We appreciate the reviewer’s useful opinion. We had shortened “Gastric cancer” to “GC” in our manuscript.
Comments 5: Line 101: space is lacking before [9]. Response 5: We thank and agree with the reviewer suggestion and add space.
Comments 6: Line 117: “Type IV” should be “type IV” Response 6: Thanks for the reviewer’s suggestion and we had corrected in line 123.
Comments 7: Line 120: “Ras–ERK” should be “RAS-ERK” Response 7: We thank and agree with the reviewer suggestion and rewrite in line 126.
Comments 8: Line 126: Should be (Figure 1/2) – provide figure 1 after mentioning it in the manuscript. Response 8: We appreciate the reviewer’s useful opinion and corrected in line 132.
Comments 9: Line 127: NF-κB should be provided with a full name. Response 9: Thanks for the reviewer’s suggestion and we had provided with a full name in line 133.
Comments 10: Line 129: What is PAI. Provide full name. Response 10: We appreciate the reviewer’s useful opinion and provide full name in line 135.
Comments 11: Line 130: TAK1, TRAF6 lacks full name the first time it appears in the manuscript. Response 11: We thank and agree with the reviewer suggestion and rewrite in line 137-139.
Comments 12: Line 133: “Akt” should be replaced with “AKT” Response 12: Thanks for the reviewer’s suggestion and we had corrected in line 140.
Comments 13: Line 137: IL – need the full name. Response 13: We thank and agree with the reviewer suggestion and rewrite in line 145.
Comments 14: Line 141, 144 – lack of space before citation [x] – multiple such errors have been detected in the manuscript. Please review and correct them accordingly. Response 14: We appreciate the reviewer’s useful opinion and corrected these errors.
Comments 15: Line 173-175: Please rewrite: “However, some studies disagreed due to their findings of a minimal increase in the incidence of colon adenoma in patients with H. pylori infection” for enhanced clarity and precision. Response 15: Thanks for the reviewer’s suggestion and we had rewritten in line 181-183.
Comments 16: Line 177: introduces the new abbreviation “colorectal cancer (CRC)” which appeared multiple times in the text before the first introduction. Please decide which one to use – the abbreviated form or the full form throughout the manuscript. Response 16: We thank and agree with the reviewer suggestion and corrected these words.
Comments 17: Line 191-193: “The authors also examined white, black, Asian/Pacific Islander, American, Indian, and Hispanic race populations and showed that H. pylori infection might affect the development of CRC and let this study result was more consolidated” Please rewrite for clarity. Response 17: Thanks for the reviewer’s suggestion and we had rewritten in line 199-202.
Comments 18: Lines 262-265: Please rewrite for clarity. Response 18: We appreciate the reviewer’s useful opinion and rewrite in line 271-275.
Comments 19: Line 310: Remove “in animal study” Response 19: We thank and agree with the reviewer suggestion and removed these words.
Comments 20: Line 336: Please rewrite from: “Wong et al. had demonstrated, via sequencing, uncovered shifts in microbial composition and ecology in individuals with CRC” to “Wong et al. demonstrated through sequencing that there are significant shifts in microbial composition and ecology in individuals with CRC” Response 20: Thanks for the reviewer’s suggestion and we had rewritten in line 349-350.
Comments 21: Line 337-341: “Additionally, research has identified specific bacteria, such as Fusobacterium nucleatum, certain strains of Escherichia coli, and Bacteroides fragilis, and their respective roles in the development of CRC. Especially, Streptococcus gallolyticus, B. fragilis, and F. nucleatum could have direct effects on the formation of colon neoplasms or cancer” please decide if you want to provide a full name or abbreviation e.g. Bacteroides fragilis or B. fragilis please unify it throughout the manuscript. Response 21: We appreciate the reviewer’s useful opinion and rewrite in line 353.
Comments 22: Line 341: Consider rewriting to “Several human shotgun metagenomic and 16S ribosomal RNA sequencing studies have been conducted to analyze the microbiota associated with colorectal cancer (CRC), using fecal and mucosal samples to characterize these microorganisms.” Response 22: Thanks for the reviewer’s suggestion and we had rewritten in line 354-357.
Comments 23: Table 1. “odd ratio” seems highlighted. Response 23: Thanks for the reviewer’s suggestion and we had rewritten in Table 1.
Comments 24: Line 379: “Since H. pylori infection was the most infectious disease in human kind.” This statement seems unfinished or oddly inserted here. Please rewrite. Response 24: We appreciate the reviewer’s useful opinion and rewrite in line 392-396.
Comments 25: The reference list seems disordered please correct and check whether the reference numbers match the content of the manuscript. Provide a reference list and the statements (including contribution statements) according to the journal requirements. Pay attention to the layout requirements of the journal - including affiliations. Response 25: We thank and agree with the reviewer suggestion and corrected our article reference.
|

Reviewer 2 Report
Comments and Suggestions for Authors The subject of this review focuses on summarizing the central points of the role of *Helicobacter pylori* (H. pylori) infection in the development of gastric cancer (GC) and colorectal neoplasms (CRC), from early to recent studies. Particularly, the association between H. pylori and CRC is still a matter of debate, making it important to provide an overview that offers valuable insights. This paper sheds light on how changes in the gut microbiota and chronic inflammation caused by H. pylori are involved, contributing new knowledge to this field. The aspect of H. pylori causing dysbiosis (imbalance) in the gut microbiota represents a relatively new perspective. However, looking at the reference list of this review, I am concerned about the large number of citations to other review papers. In general, it is not preferable for a review to cite other reviews. This is because the purpose of a review paper is usually to organize and synthesize the latest knowledge based on original research (primary articles). Citing information already summarized in another review may weaken direct access to the primary data and its reliability. There are instances where citing a review paper as a comprehensive resource to provide a historical background or overview of the field is acceptable. Additionally, this paper effectively references the results of systematic reviews. However, the prominent use of review citations in critical parts of the text is noticeable. For example, is it appropriate to cite reviews in references 4, 5, 10, 11, 19, and 36? Additional comments: In line 309, the term "Second" is unclear because there is no corresponding "First" in the preceding text. Please revise for clarity.Author Response
|
Comments 1: I am concerned about the large number of citations to other review papers. In general, it is not preferable for a review to cite other reviews. This is because the purpose of a review paper is usually to organize and synthesize the latest knowledge based on original research (primary articles). Citing information already summarized in another review may weaken direct access to the primary data and its reliability. There are instances where citing a review paper as a comprehensive resource to provide a historical background or overview of the field is acceptable. Additionally, this paper effectively references the results of systematic reviews. However, the prominent use of review citations in critical parts of the text is noticeable. For example, is it appropriate to cite reviews in references 4, 5, 10, 11, 19, and 36? |
|
Response 1: Thank you for pointing this out. We agree with this comment. Therefore, we had changed references 10, 19 and added reference 48 and 54 as your suggestion.
|
|
Comments 2: Additional comments: In line 309, the term "Second" is unclear because there is no corresponding "First" in the preceding text. Please revise for clarity.
|
|
Response 2: Agree. Thank you for your suggestion. We had rewritten this section as your suggestion.
|

Reviewer 3 Report
Comments and Suggestions for Authors
The revision of this paper calls for some important improvements:
1. Research methodology: The paper lacks a well-defined section on research methodology. It is essential to include a detailed description of the study selection criteria, the data sources used, and the statistical methods employed for the analysis.
2. Summary table of papers: A summary table should be included to summarise the main studies reviewed, including key information such as: author, year, type of study, number of participants, main findings, and conclusions.
3. Discussion of the pros and cons of H. pylori eradication: The discussion does not sufficiently address the potential advantages and disadvantages of H. pylori eradication. It would be useful to include a section that balances the benefits in terms of reduced risk of gastric and colorectal cancer with the possible disadvantages, such as increased antibiotic resistance and impact on the gut microbiota.
4. Beware of plagiarism. The work must be reworded and paraphrased in an original and accurate manner to reduce similarities with existing sources. See file.

Author Response
|
Comments 1: Research methodology: The paper lacks a well-defined section on research methodology. It is essential to include a detailed description of the study selection criteria, the data sources used, and the statistical methods employed for the analysis. |
|
Response 1: Thanks for your valuable suggestion. Because our article is defined as narrative review rather than systematic review, selection criteria, re-analysis and summary table were not presented in our manuscript. Systematically review the relevant article might be a fantastic idea for our further research. Thank you very much.
|
|
Comments 2: Summary table of papers: A summary table should be included to summarise the main studies reviewed, including key information such as: author, year, type of study, number of participants, main findings, and conclusions..
|
|
Response 2: We appreciate your valuable suggestion. As our article is categorized as a narrative review rather than a systematic review, we did not include selection criteria, re-analysis, and summary tables in our manuscript. However, conducting a systematic review of relevant articles could be a fantastic idea for our future research. Thank you for your input.
|
|
Comments 3: Discussion of the pros and cons of H. pylori eradication: The discussion does not sufficiently address the potential advantages and disadvantages of H. pylori eradication. It would be useful to include a section that balances the benefits in terms of reduced risk of gastric and colorectal cancer with the possible disadvantages, such as increased antibiotic resistance and impact on the gut microbiota. Response 3: Thanks for your valuable suggestion. We had added potential advantages and disadvantages of H. pylori eradication in line 97-102.
Comments 4: Beware of plagiarism. The work must be reworded and paraphrased in an original and accurate manner to reduce similarities with existing sources. See file. Response 4: Thank you for pointing this out. We agree with this comment. Therefore, we had rewritten our article and reduce similarities with existing sources.
|

Reviewer 4 Report
Comments and Suggestions for Authors
This is interesting topic as H. pylori infection is prevalent worldwide. However, manuscript has serious flaws and needs to be improved in order for it to be acceptable for publication. For instance, space is missing before breckets many times, English language and style could be improved, more literature on this matter must be included in the main part of the review, reference for table 1 is missing etc. I believe this article could add to the body of literature in this field, if the authors improved the presented manuscript.
Comments on the Quality of English LanguageShould be improved
Author Response
|
Comments 1: However, manuscript has serious flaws and needs to be improved in order for it to be acceptable for publication. For instance, space is missing before breckets many times. |
|
Response 1: Thanks for your valuable suggestion. We had added space before breckets in our manuscript. Thank you very much.
|
|
Comments 2: English language and style could be improved, more literature on this matter must be included in the main part of the review, reference for table 1 is missing etc. |
|
Response 2: We appreciate your valuable suggestion. We had added reference in Table 1 and improved out English language and style. Thank you for your input.
|

Round 2
Reviewer 3 Report
Comments and Suggestions for Authors
Despite the requested improvements, the manuscript still does not meet the scientific requirements for publication. The methodology section remains insufficient; there is a lack of clear description of the selection criteria and methods of analysis. This compromises the transparency and replicability of the work. The summary table, essential for a clear review of studies, is still lacking (author, year, study type, sample, main results, conclusions).
Discussion of the pros and cons of H. pylori eradication remains limited and unbalanced. Risks such as antibiotic resistance and impact on the microbiota are not adequately explored.
Problems of Originality: Problems of similarity with existing sources persist, indicating an insufficient level of reworking and originality (see mdpi report).